

# Errors and improvements in the use of archived meteorological data for chemical transport modeling

Karen Yu[1], Christoph A. Keller[2,3], Daniel J. Jacob[1,4], Andrea M. Molod[3], Sebastian D. Eastham[1], and Michael S. Long[1]

[1]School of Engineering and Applied Sciences, Harvard University, Cambridge, MA, USA
[2]Universities Space Research Association, Columbia, MD, USA
[3]NASA Goddard Space Flight Center, Greenbelt, MD, USA
[4]Department of Earth and Planetary Sciences, Harvard University, Cambridge, MA, USA

*Correspondence to:* kyu@seas.harvard.edu

**Abstract.** Global simulations of atmospheric chemistry are generally conducted with off-line chemical transport models (CTMs) driven by archived meteorological data from general circulation models (GCMs). The off-line approach has advantages of simplicity and expediency, but incurs errors due, in part, to temporal averaging in the meteorological archive and the inability to reproduce the GCM transport algorithms exactly. The CTM simulation is also often conducted at coarser grid resolution than the parent GCM. Here we investigate this cascade of CTM errors by using $^{222}$Rn-$^{210}$Pb-$^{7}$Be chemical tracer simulations off-line in the GEOS-Chem CTM at rectilinear $0.25° \times 0.3125°$ ($\approx 25$ km) and $2° \times 2.5°$ ($\approx 200$ km) resolutions, and on-line in the parent GEOS-5 GCM at cubed-sphere c360 ($\approx 25$ km) and c48 ($\approx 200$ km) horizontal resolutions. The c360 GEOS-5 GCM meteorological archive, updated every 3 hours and remapped to $0.25° \times 0.3125°$, is the standard operational product generated by the NASA Global Modeling and Assimilation Office (GMAO) and used as input by GEOS-Chem. We find that the GEOS-Chem $^{222}$Rn simulation at native $0.25° \times 0.3125°$ resolution is affected by vertical transport errors of up to 20% relative to the GEOS-5 c360 on-line simulation, in part due to loss of transient organized vertical motions in the GCM (resolved convection) that are temporally averaged out in the 3-hour meteorological archive. There is also significant error caused by operational remapping of the meteorological archive from cubed-sphere to rectilinear grid. Decreasing the GEOS-Chem resolution from $0.25° \times 0.3125°$ to $2° \times 2.5°$ induces further weakening of vertical transport as transient vertical motions are averaged out spatially as well as temporally. The resulting $^{222}$Rn concentrations simulated by the coarse-resolution GEOS-Chem are overestimated by up to 40% in surface air relative to the on-line c360 simulations, and underestimated by up to 40% in the upper troposphere, while the tropospheric lifetimes of $^{210}$Pb and $^{7}$Be against aerosol deposition are affected by 5-10%. The lost vertical transport in the coarse-resolution GEOS-Chem simulation can be partly restored by re-computing the convective mass fluxes at the appropriate resolution to replace the archived convective mass fluxes, and by correcting for bias in spatial averaging of boundary layer mixing depths.



## 1  Introduction

Accurate simulation of transport is crucial for global models of atmospheric composition. Transport information is provided by output from general circulation models (GCMs) that solve the conservation equations for air mass, momentum, heat, and water, and may assimilate meteorological observations to reproduce a specific period. GCMs compute grid-resolved winds,
subgrid turbulence, and convection properties that determine the transport of chemical species through the corresponding continuity equations (Brasseur and Jacob, 2017). These equations can be solved "on-line" as part of the GCM or "off-line" by using archived winds and turbulence statistics to drive a separate chemical transport model (CTM). The off-line approach has advantages of simplicity and economy, but it introduces differences due, in large part, to temporal (and sometimes spatial) averaging in the meteorological archive. Since the CTM aims to replicate the original transport of the GCM, any deviation from
the GCM transport can be viewed as an error. Here we use chemical tracers to investigate the cascade of errors involved in successively degrading a global on-line simulation with high spatial resolution through various stages to an off-line simulation with coarse horizontal spatial resolution and coarse temporal resolution of input data.

Whether on-line or off-line, a model of atmospheric composition computes the concentrations of atmospheric species by solving the relevant chemical continuity (mass conservation) equations. In an Eulerian (fixed frame of reference) framework,

$$\frac{\partial \rho_i}{\partial t} = -\nabla \cdot (\rho_i \mathbf{v}) + \nabla \cdot \left( \mathbf{K} \rho_a \nabla \frac{\rho_i}{\rho_a} \right) + P_i - L_i \tag{1}$$

Here $\rho_i$ is the mass density of species $i$, $\rho_a$ is the air density, $\mathbf{v}$ is the wind vector, $\nabla \cdot (\rho_i \mathbf{v})$ represents the advection term (flux divergence), and $P_i - L_i$ accounts for local production and loss as from chemical reactions. Small-scale turbulent transport is parameterized in equation (1) as an eddy diffusion term where $\mathbf{K}$ is an eddy diffusivity tensor. Additional parameterizations are applied for convection, which is sub-grid on the horizontal scale but organized (non-local) on the vertical scale. Unlike
the Navier-Stokes conservation equation for momentum, where non-linear dependence on momentum introduces chaos in the solution, the chemical continuity equation has stable solutions when $\mathbf{v}$ and $\mathbf{K}$ are specified. This is an important motivation for decoupling the CTM from the GCM, and to use archives of $\mathbf{v}$, $\mathbf{K}$, and convection diagnostics to drive the off-line transport.

A GCM typically uses a time step of the order of minutes to integrate the conservation equations for atmospheric dynamics. In an on-line model, the chemical continuity equations can be integrated using updated winds on the same time step. But
archiving winds at that resolution for off-line CTM applications is impractical in terms of data storage. Instead, meteorological archives for use in CTMs are typically available as temporal averages every few hours, losing information on eddy motions at shorter time scales that might affect chemical transport. Rasch et al. (1997) found that 6-h archiving of GCM meteorological fields did not induce significant off-line chemical transport error but 24-h archiving did. Dentener et al. (1999) confirmed that CTMs using meteorology archived at 6-h intervals could reproduce the transport of the originating GCM. These older GCMs
used grid resolutions of hundreds of km, whereas current GCMs use tens of km. The error from temporal averaging increases with increasing grid resolution, particularly as the GCM becomes fine enough to partly resolve convective scales (Grell et al., 2004).

Deep convection is of concern for off-line CTM applications. Vertical convective motions driven by buoyancy are subgrid on the horizontal scale but organized on the vertical scale, transporting air across several vertical model levels in a single time





step. Deep convection enables the transport of short-lived species to high altitudes and scavenges water-soluble species (such as aerosol particles) in the cloud updrafts. Convective parameterizations used in GCMs diagnose cloud updrafts, downdrafts, detrainment/entrainment, and compensatory large-scale subsidence on the grid scale (Brasseur and Jacob, 2017). It is common practice to use temporally averaged convective mass fluxes from the GCM archives to drive off-line models but the exact timing

of events is lost. A compounding problem is that current GCMs have sufficiently fine resolution to partly resolve convective systems on the grid scale, and so the parameterized convection is suppressed in scale-aware schemes (Grell and Freitas, 2014; Molod et al., 2015). Typical convective systems persist for less than an hour, and the corresponding advective transport is averaged out in a multi-hour wind archive.

Spatial averaging of the meteorological archive is yet another concern. CTM simulations of oxidant-aerosol chemistry and/or

10 aerosol microphysics may require over 100 coupled species. The computational costs are large. A way to reduce costs is to degrade spatial resolution. It is thus common practice in CTM applications to average the GCM meteorological fields onto coarser grids for input to the CTM, and operate the CTM at that coarser resolution. But the averaging may introduce transport biases. For example, vertical velocity and eddy fluxes resolved at the native GCM resolution may be lost by averaging to the coarser grid.

Here we examine how off-line archiving of GCM meteorological data, including temporal and spatial averaging, affects the simulation of transport in the GEOS-Chem CTM, and we recommend some corrections for these errors. We use for this purpose the $^{222}$Rn-$^{210}$Pb-$^7$Be tracer suite, which provides a standard basis for evaluating transport and aerosol scavenging in CTMs (Jacob et al., 1997; Liu et al., 2001; Considine et al., 2005; Zhang et al., 2017). The GEOS-Chem CTM, originally described by Bey et al. (2001), is an open-source global model of atmospheric composition used by a large research community for a

wide range of applications. The experiments described here rely on assimilated meteorological data archived every 3 hours from the NASA Goddard Earth Observation System (GEOS) Data Assimilation System (DAS, Rienecker et al., 2011; Gelaro et al., 2017) on a cubed-sphere grid, interpolated to $0.25° \times 0.3125°$ ($\approx$25 km) horizontal resolution. The GEOS DAS uses the underlying GCM described in Rienecker et al. (2008) and Molod et al. (2015). GEOS-Chem CTM simulations can be conducted at that native resolution, but global simulations generally use degraded $2° \times 2.5°$ ($\approx$200 km) or $4° \times 5°$ ($\approx$400

25 km) resolution for computational expediency. The GEOS-Chem chemical module (solving $\frac{\partial \rho_i}{\partial t} = P_i - L_i$) has recently been integrated within the GEOS GCM so that simulations with detailed chemistry can be conducted either on-line or off-line using the exact same module (Long et al., 2015).

## 2 Model Descriptions

### 2.1 GEOS-5 GCM

The Goddard Earth Observing System Model, Version 5 (GEOS-5) is a GCM developed by the NASA Global Modeling and Analysis Office (GMAO) (Rienecker et al., 2008; Molod et al., 2015). Advection is driven by the finite volume dynamical core of Putman and Lin (2007), which uses substepping to ensure the Courant number does not exceed unity. Boundary layer mixing is based on the non-local scheme of Lock et al. (2000) and the Richardson-number-based scheme of Louis et al. (1982).





The convective parameterization is the Relaxed Arakawa-Schubert (RAS) scheme (Moorthi and Suarez, 1992) with a scheme for generation and re-evaporation of precipitation (Bacmeister et al., 2006). RAS computes the effect of multiple individual cloud plumes released sequentially, using a resolution-dependent stochastic trigger function (Bacmeister and Stephens, 2011). GEOS-5 has 72 vertical levels up to 0.01 hPa on a hybrid eta (sigma-pressure) grid. The horizontal grid is cubed-sphere
(Putman and Lin, 2007) and can operate at a range of resolutions. The integration of the model equations on the cubed-sphere grid eliminates the problem of large Courant numbers near the poles and permits straightforward domain decomposition for distributed-memory environments.

We use here the operational GEOS-5 product (https://gmao.gsfc.nasa.gov/GMAO_products/NRT_products.php) generated at a cubed-sphere c360 ($\approx$25 km) horizontal resolution. The data combines the GEOS-5 GCM with observations using a hybrid
ensemble Kalman filter 3-dimensional variational (3DVAR-hybrid) system (Rienecker et al., 2012). The internal GCM time step for advection and convection is 7.5 minutes. Output from the c360 simulation is mapped onto a $0.25° \times 0.3125°$ latitude $\times$ longitude rectilinear grid to produce the GEOS forward processing (GEOS-FP) archive released operationally by GMAO. The archived data relevant to CTM transport and scavenging include 3-D winds, convective mass fluxes, and precipitation fields; and 2-D surface pressures and boundary layer mixing depths. The 3-D data are archived as 3-hour averages and the 2-D data
as 1-hour averages.

GEOS operational meteorological products have been widely used for off-line CTM applications including by the University of Maryland (UMD) CTM (Allen et al., 1996), GEOS-Chem (Bey et al., 2001), Global Modeling Initiative (GMI) CTM (Douglass et al., 1999, 2004), Integrated Massively Parallel Atmospheric Chemical Transport (IMPACT) model Rotman et al. (2004), MOZART (Emmons et al., 2010), CAM-Chem (Lamarque et al., 2012), and the GEOS-CTM (Kouatchou et al., 2015).
Chemical transport simulations can also be performed on-line within the GEOS-5 GCM (Colarco et al., 2010; Oman and Douglass, 2014; Long et al., 2015; Strode et al., 2015; Li et al., 2016).

## 2.2   GEOS-Chem CTM

We use two versions of GEOS-Chem: the standard version 11-01 released in February 2017
(http://wiki.seas.harvard.edu/geos-chem/index.php/GEOS-Chem_v11-01) and a beta high-performance version (GCHP) de-
25 signed for massively parallel computing environments (http://wiki.seas.harvard.edu/geos-chem/index.php/GEOS-Chem_HP).
The standard GEOS-Chem operates on the rectilinear grid from the GEOS-5 archive, while GCHP operates on the GEOS-5 cubed-sphere grid. Both versions use the same archived meteorological data and modules except for advection. GCHP uses an off-line version of the same Putman and Lin (2007) dynamical core as GEOS-5, while the standard GEOS-Chem uses a dynamical core developed for off-line applications on a rectilinear grid (Lin and Rood, 1996). The Lin and Rood (1996)
scheme averages winds, surface pressures, and mixing ratios over the highest two latitude bands to compensate for the polar singularity. Vertical advection is computed in both cases from the change in surface pressure, but vertical advection in the standard rectilinear GEOS-Chem is lower-order than in GEOS-5 and GCHP. In the standard rectilinear GEOS-Chem, a pressure fixer (Horowitz et al., 2003) is used to correct for inconsistencies between horizontal wind divergence and pressure tendencies



resulting from the temporal averaging in the meteorological archive, whereas GCHP enforces mass consistency by applying a global scaling factor.

Convective transport in GEOS-Chem is simulated with a single-plume scheme using the archived 3-hour net updraft and detrainment convective mass fluxes summing over all RAS plumes within a given grid column (Wu et al., 2007). Although

GEOS-Chem reproduces the bulk 3-h convective transport in the GEOS-5 GCM, the precise timing and interactions between RAS convective plumes is not resolved since that information is not in the archive.

Boundary layer mixing in GEOS-Chem uses the non-local parameterization of Holtslag and Boville (1993) adapted for GEOS-Chem by Lin and McElroy (2010). It draws on the archived GEOS-5 mixing depths, temperature, latent and sensible heat fluxes, and specific humidity. The mixing depths in the GEOS-5 archive are a diagnostic quantity, and thus boundary layer

mixing in GEOS-Chem may differ from that in the GEOS-5 GCM.

GEOS-Chem applications typically use the native horizontal resolution of the GEOS products for nested simulations over continental-scale domains (Wang et al., 2004; Chen et al., 2009; Kim et al., 2015), but a coarser $2° \times 2.5°$ or $4° \times 5°$ horizontal resolution for global simulations. The $2° \times 2.5°$ and $4° \times 5°$ meteorological archives are generated by averaging the original GEOS-5 archive over the corresponding grid. As part of this work, we developed a capability to conduct global GEOS-Chem

simulations for passive tracers at native $0.25° \times 0.3125°$ horizontal resolution. This allows us to separate the contributions of off-line archiving and degraded resolution to model errors.

## 3   Simulation Ensemble

### 3.1   The $^{222}$Rn-$^{210}$Pb-$^{7}$Be system

The natural tracer suite $^{222}$Rn-$^{210}$Pb-$^{7}$Be provides a standard test of vertical transport and scavenging in global models, with

capability to compare to observations (Liu et al., 2001). $^{222}$Rn is emitted ubiquitously by soils. Its sole sink is radioactive decay to $^{210}$Pb with a half-life of 3.8 days, making it a sensitive tracer for vertical transport in the troposphere (Liu et al., 1984; Jacob and Prather, 1990; Jacob et al., 1997; Allen et al., 1996). $^{210}$Pb (half-life 22.3 years) attaches to aerosol particles and provides a diagnostic for aerosol lifetime against deposition (Balkanski et al., 1993). $^{7}$Be (half-life 53.3 days) is produced in the upper troposphere and lower stratosphere from the interaction of cosmic rays with atmospheric oxygen and nitrogen (Lal and Peters,

1967), and attaches to aerosol particles and is removed by deposition in the same way as $^{210}$Pb. The high-altitude source of $^{7}$Be complements $^{210}$Pb by testing the model representation of subsidence and stratosphere-troposphere exchange (Koch et al., 1996; Considine et al., 2005).

Here we use the $^{222}$Rn-$^{210}$Pb-$^{7}$Be simulation originally developed for GEOS-Chem by Liu et al. (2001). $^{222}$Rn is emitted uniformly from land excluding ice at a rate of 1.0 atoms cm$^{-2}$ s$^{-1}$ under non-freezing conditions and 0.3 atoms cm$^{-2}$ s$^{-1}$

under freezing conditions. The $^{7}$Be source function depends only on altitude and latitude. $^{210}$Pb and $^{7}$Be are removed by aerosol wet and dry deposition, in addition to radioactive decay (negligible for $^{210}$Pb). Dry deposition is a minor sink. Wet deposition includes scavenging in convective updrafts following the approach of Balkanski et al. (1993), as well as first-order in-cloud and below-cloud scavenging for anvil and large-scale (grid-resolved) precipitation following the approach of Giorgi





and Chameides (1986). Aerosol can be released below cloud if precipitation evaporates. The scavenging parameterizations of Liu et al. (2001) are intended to be applicable to GEOS-Chem at all resolutions, because convective mass fluxes (from the GEOS-5 archive) do not change with resolution and because first-order rainout/washout assumes precipitating fractions of gridboxes that are set in all cases by a fixed rate of conversion of cloudwater to precipitation (Giorgi and Chameides, 1986).

5   A recent study by Zhang et al. (2017) uses GEOS-Chem with an updated $^{222}$Rn source function to evaluate with observations worldwide. Here our focus is not on comparison to observations but on the effect of CTM model differences relative to a reference simulation. Since the averaging/remapping of meteorological fields in the CTM represents a degradation of the information from the reference simulation, we view for our purpose the reference simulation as the "truth" against which the different CTM simulations can be compared.

## 3.2   Simulations performed

We conducted a number of on-line and off-line $^{222}$Rn-$^{210}$Pb-$^{7}$Be simulations for different spatial resolutions and configurations, as illustrated in Figure 1 and explained below. All simulations were conducted for two months, starting from zero concentrations on June 1, 2013. We report and compare monthly mean results for July 2013 after a one-month (June) spin-up. We limited the analysis to one month because of computational and storage requirements for the high-resolution simulations,

and with the expectation that one month in northern hemisphere summer is a sufficient time window to diagnose systematic differences in vertical tropospheric transport as revealed by the $^{222}$Rn-$^{210}$Pb-$^{7}$Be system. The zero initialization allows for sensitive analysis of differences but implies that concentrations are not in steady state and should not be compared to observations, in particular for $^{210}$Pb and $^{7}$Be. For example, observed and steady-state GEOS-Chem $^{210}$Pb concentrations in the stratosphere are higher than in the troposphere (Liu et al., 2001) but in our simulations they are much lower. Stratospheric concentrations

of $^{222}$Rn and $^{210}$Pb should not be compared across simulations since injection of $^{222}$Rn to the stratosphere may be driven by sporadic deep convection (Lambert et al., 1990). The stratosphere is not discussed in what follows.

Simulation 1 is conducted on-line using a version of GEOS-5 similar to the one used in Molod et al. (2015) at c360 resolution. For $^{222}$Rn, it is the reference simulation for our purposes; all other simulations in Figure 1 (blue boxes) degrade successively some aspect of that reference simulation. This is not the case for $^{210}$Pb and $^{7}$Be, because the scheme for aerosol wet scavenging

is less advanced than in GEOS-Chem. In particular, aerosol scavenging in the GCM is not coupled to subgrid transport in deep convective updrafts, and this can severely overestimate the transport of aerosols to the upper troposphere (Balkanski et al., 1993). Thus we do not show $^{210}$Pb and $^{7}$Be results from the on-line simulations.

Simulation 2 is also conducted on-line at c360 resolution but with the bulk convective algorithm of GEOS-Chem and 3-h averaged convective mass fluxes from Simulation 1. This allows us to separately examine the effect of using archived fields on convection and advection. Simulation 2 is used to generate the meteorological archive (3-h for winds and convective mass

fluxes, 1-h for mixing depths) for the off-line GEOS-Chem simulations. This off-line archive mimics the operational GEOS-FP archive by using the same temporal averaging windows and remapping the cubed-sphere meteorological data to a $0.25° \times 0.3125°$ rectilinear grid.





Simulation 3 is the standard off-line high-resolution GEOS-Chem on a $0.25° \times 0.3125°$ rectilinear grid. It uses an archive of winds, mixing depths, and convective mass fluxes generated from Simulation 2 and mimicking the GEOS-5 operational product. Errors in the $^{222}$Rn simulation compared to Simulation 2 include the temporal averaging of winds and mixing depths, the remapping of the meteorological archive to a rectilinear grid, and the use of a lower-order advection core and a different

boundary layer mixing scheme. For $^{210}$Pb and $^{7}$Be, Simulation 3 represents our best-case reference simulation.

Simulation 4 is the standard off-line coarse-resolution GEOS-Chem on a $2° \times 2.5°$ grid. It uses the off-line meteorological archive from Simulation 2 but degraded to $2° \times 2.5°$ resolution. Comparison with Simulation 3 shows the error from degraded horizontal resolution. Comparison to Simulation 1 (for $^{222}$Rn) shows the compounded errors in going from the original on-line GEOS-5 simulation to the off-line, coarse-resolution simulation.

Although global GEOS-Chem simulations may be conducted at coarse $2° \times 2.5°$ or $4° \times 5°$ resolution, they use driving meteorological fields generated from the original GEOS-5 simulation at c360 ($\approx 0.25° \times 0.3125°$) resolution. This is an important distinction from a simulation that would be driven by a coarser meteorological model. To investigate that effect, we also conducted an on-line Simulation 5 using GEOS-5 meteorology at c48 resolution ($\approx 2° \times 2.5°$). We then used the meteorological data from Simulation 5, archived on the cubed-sphere grid, to drive an off-line c48 simulation using the high-performance

version of GEOS-Chem on that grid (Simulation 6) and an off-line $2° \times 2.5°$ simulation using the standard GEOS-Chem on a rectilinear grid (Simulation 7). Comparison of Simulations 6 and 7 diagnoses the error from remapping of the meteorological archive from its native cubed-sphere to a rectilinear grid. Except for temporal averaging, the off-line archive for Simulation 6 is fully consistent with the c48 on-line Simulation 5 (no remapping).

Together, Simulations 1-7 allow us to examine and isolate different sources of error in simulations of chemical transport

including meteorological grid resolution, off-line meteorological archiving (temporal averaging), remapping of the meteorological archive, spatial degradation of that archive, and differences between off-line and on-line transport schemes. Salient results are discussed in the next section. We use monthly-average zonal mean profiles vs. altitude and latitude as our comparison metric, following standard practice for $^{222}$Rn model intercomparisons (Jacob et al., 1997). Another comparison metric for $^{210}$Pb and $^{7}$Be is the global tropospheric lifetime against deposition (Liu et al., 2001). Throughout this paper, we refer to

"archiving" as the temporal averaging of meteorological fields for use in off-line simulations, "remapping" as the cubed-sphere to rectilinear transformation of these fields, and "spatial averaging" as the further degradation of these fields from a fine to a coarse off-line grid.

## 4 Simulation results

Figure 2 (left panel) shows the zonal July mean profile of $^{222}$Rn concentrations as a function of latitude and altitude from

30 Simulation 1. The latitudinal distribution reflects the continental source. The $^{222}$Rn lifetime is much shorter than the vertical mixing time of the troposphere ($\sim$1 month), resulting in strong vertical gradients. The zonal mean concentration patterns are typical of those found in models (Jacob et al., 1997).





## 4.1 Errors from use of off-line convection scheme

The middle panel of Figure 2 shows the percentage differences in zonal mean $^{222}$Rn profiles between Simulation 2 (off-line GEOS-Chem convection scheme) and Simulation 1. Simulation 2 has up to 10% higher $^{222}$Rn concentrations in the equatorial lower troposphere and up to 7% lower $^{222}$Rn concentrations in the mid to upper troposphere. The combination of using the

GEOS-Chem convection scheme and using temporally averaged convective mass fluxes results in slightly reduced vertical transport compared to the original GEOS-5 convection.

The GEOS-Chem convection scheme operates as a single convective plume in each grid column on the basis of the 3-hour archive of GEOS-5 convective updraft and detrainment data. We find in a sensitivity simulation that using 15-min or 3-hour averages of convective mass fluxes makes no significant difference. Thus the differences in Figure 2 arise mainly from

the bulk convective transport scheme used in GEOS-Chem, which simplifies the RAS ensemble-plume parameterization to a single plume. One explanation for why a multi-plume parameterization might produce a different transport pattern is that each sequential plume acts on a different concentration gradient that has been modified by the previous plume. A tall plume followed by a series of short plumes will transport more tracer higher than a series of short plumes followed by a tall plume.

## 4.2 Errors from off-line vs. on-line simulation

The right panel of Figure 2 shows the percentage differences between Simulation 3 (off-line) and Simulation 2 (on-line). The off-line simulation has higher $^{222}$Rn concentrations in the mid-troposphere (700-500 hPa) and lower concentrations in the upper troposphere (above 500 hPa), with some differences exceeding 20%. Encapsulated in this comparison are the effects of remapping the archived meteorological fields to rectilinear grid, using a different advection scheme, using a different boundary layer mixing scheme, and using 3-hourly averaged wind fields. Large differences in polar grid cells may reflect averaging

of concentrations in the polar latitudes in the rectilinear advection scheme (Lin and Rood, 1996) as well as the transition to semi-Lagrangian advection when the Courant number exceeds unity.

To better understand the contributions from different sources of error in the off-line simulation, we examine Simulations 5-7, which show a similar transition from on-line to off-line, but starting at c48 resolution in the GEOS-5 model and with the intermediate addition of a c48 off-line GCHP simulation (custom cubed-sphere archive, no remapping). In this way we can

diagnose the effects of using archived winds separately from the effects of the advection core and remapping error associated with conversion to a rectilinear (here $2° \times 2.5°$) grid. The left panel of Figure 3 shows the zonal mean $^{222}$Rn concentrations for c48 resolution, which have a similar pattern to the c360 results. The middle and right panels of Figure 3 separate out additively the contributions from off-line archiving (Simulation 6 vs. Simulation 5) and remapping and transport scheme (Simulation 7 vs. Simulation 6). Off-line archiving results in overall weaker vertical transport, as might be expected from transient motions averaged out in the meteorological archive. The bias is about 5% in surface air but can exceed 20% in the upper troposphere.

There is still a bias over Antarctica even though the off-line cubed-sphere geometry does not have a polar singularity; this may reflect the cumulative effect of meridional transport differences affecting a region particularly remote from sources. The combination of remapping to the rectilinear grid and using the Lin and Rood (1996) advection scheme (right panel) also incurs





differences of about 5% in surface air and up to 20% in the upper troposphere. We would expect to see larger remapping errors associated with smaller grids, especially over the polar regions. The errors from Simulations 6 and 7 compound for surface air but tend to cancel in the upper troposphere.

### 4.3 Errors from grid resolution

Figure 4 shows the effect of grid resolution on zonal mean $^{222}$Rn profiles in GEOS-5 (on-line, with resolution affecting meteorology) and GEOS-Chem (off-line, using the same meteorological archive in both cases). The left panel shows the percentage difference between Simulation 5 (on-line c48) and Simulation 2 (on-line c360). The c360 meteorological simulation has sufficient spatial resolution to resolve large convective systems, and thus has much less parameterized convection than the c48 simulation. The higher $^{222}$Rn concentrations in the tropical upper troposphere in c48, and lower concentrations in

the extratropical upper troposphere, most likely reflects differences in vertical transport properties between the resolved and parameterized convective formulations. Thus the higher concentrations in the tropical mid-upper troposphere at c48 can be attributed to the stronger action of the convective parameterization that is not "picked up" by the resolved transport. The lower concentrations near the tropopause can be attributed to the known insufficiency of convective transport for reaching that level (Ott et al., 2009). The lower concentrations in the extratropical upper troposphere may be due to inability to diagnose organized

vertical motion as convection. The higher concentrations over Antarctica at c48 may be due to numerical diffusion during the slow long-range meridional transport from lower latitudes (Eastham and Jacob, 2017).

The effect of degrading model resolution has different effects in GEOS-Chem, where the coarse $2° \times 2.5°$ simulation uses the same $0.25° \times 0.3125°$ meteorological archive as the high-resolution simulation but with spatial averaging of the meteorological fields. The coarse simulation results in decreased vertical transport to the upper troposphere at all latitudes, with maximum

effect (up to 40%) in the subsiding subtropics. This may be simply explained by the averaging out of vertical eddy motions on the coarser grid, including organized vertical motions across multiple levels that the on-line c48 simulation would simulate with stronger parameterized convection. This systematic bias in the coarse-resolution GEOS-Chem may thus be correctable through the addition of convective motions. We explore this idea in the next section.

### 4.4 Errors from grid resolution for $^{210}$Pb and $^{7}$Be

Figure 5 (top panels) shows the zonal mean concentrations of $^{210}$Pb and $^{7}$Be from the GEOS-Chem simulation at $0.25° \times 0.3125°$ (Simulation 3). The $^{210}$Pb distribution is shifted to higher altitudes relative to $^{222}$Rn, reflecting the effect of scavenging in the lower troposphere. $^{7}$Be shows preferential subsidence in the dry subtropics and is depleted in the lower troposphere by scavenging. $^{7}$Be concentrations are low throughout the tropical troposphere due to dominant upwelling of $^{7}$Be-depleted surface air.

Mean tropospheric lifetimes against deposition in the $0.25° \times 0.3125°$ simulation are 6.7 days for $^{210}$Pb and 17 days for $^{7}$Be. Liu et al. (2001) previously inferred $^{210}$Pb and $^{7}$Be tropospheric residence times of 9 and 21 days respectively from an earlier version of GEOS-Chem evaluated with observations. A more recent evaluation by Zhang et al. (2017), using GEOS-





Chem version 11-01 and an updated $^{222}$Rn source, finds that a residence time for $^{210}$Pb of $7 \pm 1$ days better matches the observational constraints..

The bottom panels show the effects of degrading the GEOS-Chem resolution to $2° \times 2.5°$. Overall the results are consistent with our previous finding for $^{222}$Rn that degrading the resolution weakens vertical transport. Tropospheric lifetimes decrease to

6.2 days for $^{210}$Pb and increase to 18 days for $^{7}$Be, consistent with the shifts in vertical distribution to the lower troposphere for $^{210}$Pb and to the upper troposphere for $^{7}$Be. $^{210}$Pb concentrations are higher in the tropical upper troposphere in the $2° \times 2.5°$ simulation, likely due to differences in wet scavenging.

## 5   Correcting errors in off-line simulations

Our work has shown how a cascade of errors is introduced in model transport of chemical tracers when using off-line mete-

orological archives (as opposed to on-line simulation) and when degrading the spatial resolution of these archives for computational expediency. The compounding effect is illustrated in Figure 6 (left panel), which compares the zonal mean $^{222}$Rn concentration profiles in the off-line $2° \times 2.5°$ configuration of GEOS-Chem to the on-line GEOS-5 simulation at c360 ($\approx$25 km) resolution. Concentrations are typically biased high by  20% at the surface, and biased low by  40% in the upper troposphere. We now examine how some of these errors can be alleviated.

We can categorize the errors as resulting from four different sources: (1) differences in transport algorithms between the off-line and on-line model (advection scheme, boundary layer mixing, convective parameterization); (2) remapping of the meteorological archive (as here from a cubed-sphere to a rectilinear grid); (3) temporal averaging in the meteorological archive (causing loss of eddy motions, including grid-resolved organized convection, and requiring a pressure fixer to correct horizontal winds); and (4) spatial degradation of the meteorological archive (causing further loss of eddy motions). Our work presented

in Section 4 shows that all of these general sources of error are important, and addressing some of them requires improvement of the on-line archive. For example, an obvious improvement in our case would be for the GEOS-5 meteorological archive to be available on the native cubed-sphere grid rather than remapped to a rectilinear grid. Increasing the temporal frequency of archiving would be another obvious improvement.

Here we examine the feasibility of restoring the organized vertical motions lost in the temporal averaging of the meteorolog-

ical archive or in the spatial averaging for coarse-resolution GEOS-Chem simulations. Charlton-Perez et al. (2009) previously found that vertical motions in a large-eddy simulation at 200×200 m resolution could be preserved at coarser resolution by an eddy accumulation method where upward and downward vertical winds are averaged separately. We implemented this method by taking the archived pressure velocity ($\omega$) from the native $0.25° \times 0.3125°$ meteorological archive and separately averaging the upwards and downwards values onto the coarse-resolution $2° \times 2.5°$ grid. We then compared these values to the value

computed by GEOS-Chem on the $2° \times 2.5°$ grid, took the difference as the component of vertical advection lost due to spatial degradation, and applied this difference as a vertical mass exchange velocity (i.e., eddy diffusion) between adjacent cells. We found that this made negligible change to the $^{222}$Rn simulation, implying that the transport error on our scales is due more to loss of organized convective motion than to loss of small-scale eddies.





Figure 7 shows the mean July 2013 convective mass fluxes from the c360 and c48 GEOS-5 simulations, globally as vertical profiles (left panel) and at 500 hPa as a function of latitude (right panel). Convective mass fluxes are highest just above cloud base in the lower troposphere, and highest in the northern tropics (ITCZ). The global convective mass flux is 24% weaker in the c360 than in the c48 simulation. In the c360 simulation, parameterized convection is less needed because the organized

convective motions are partly resolved. Similar to the eddy motions, much of this resolved convective motion is lost when winds produced by a high-resolution GCM are first temporally averaged in a 3-h archive, then spatially averaged to a coarse grid.

A possible way to compensate for this lost convective motion in the meteorological archive is to increase parameterized convection in the off-line CTM. A simple approach would be to increase the archived convective mass fluxes by an adjustable

factor, but this assumes that the archived fluxes are co-located with the lost convection. A more physical approach is to recompute the convective mass fluxes on the fly in the off-line CTM simulation by using the same convective parameterization (here RAS) as in the parent GCM and applied to the meteorological archive with the scale-aware settings configured for the CTM resolution. This approach incurs little additional computational cost, because computations associated with the hydrological cycle in RAS are not performed. It may still underestimate the GCM convection, because the archived meteorological fields used by

the CTM are convectively relaxed temporal averages, but this can be corrected by adjusting the convective parameterization settings.

We implemented the GEOS-5 RAS scheme within GEOS-Chem in lieu of the archived convective mass fluxes, taking as input water vapor, temperature, mixing depth, and surface pressure fields from the archived meteorological data. The RAS scheme outputs the convective air mass flux and detrainment flux at every dynamic time-step. We then used these fluxes to drive

convective transport in GEOS-Chem, retaining the GEOS-Chem convective algorithm for consistent treatment of scavenging. Off-line $2° \times 2.5°$ GEOS-Chem simulations were conducted in this manner using both c360 and c48 meteorological fields.

Figure 7 shows the resulting global mean convective mass fluxes produced by GEOS-Chem at $2° \times 2.5°$ resolution using RAS to compute the convective mass flux based on archived meteorological data from the c360 and c48 meteorological archives. The vertical and latitudinal distributions closely match those computed in GEOS-5 (Simulations 1 and 5, respectively). With

25 c360 meteorology, the RAS-computed convective mass fluxes in GEOS-Chem are 30% higher globally than in GEOS-5 at that resolution (solid and dashed red lines), responding as desired to the scale-aware settings corresponding to the coarser resolution of the CTM. With c48 meteorology, the RAS-computed convective mass fluxes in GEOS-Chem are 35% weaker globally than in GEOS-5 (solid and dashed blue lines); here the CTM has the same spatial resolution as the GCM, and the weaker convection is expected from temporal averaging of the meteorological archive as discussed above.

We can increase convection produced by RAS by applying a temperature perturbation at the surface. At the scale of global CTMs (hundreds of km), there is substantial subgrid variability of moisture and temperature, with convection occurring preferentially over the more buoyant parts of the grid cell. Using moisture and temperature fields averaged over these large grid cells, as well as over time, will result in RAS underestimating convection. Therefore, we add a temperature perturbation proportional to the vertical temperature gradient at the surface (applied only when surface temperature is greater than air temperature) to

generate increased thermodynamic instability, an approach that is also used in the GEOS-5 GCM operating at coarse resolu-





tions. We set a limit of 3.0 K as the maximum allowable temperature perturbation. This leads to convective mass fluxes that are 2.5 times as much as the archived values.

Figure 6 (middle panel) shows the effect of including RAS in GEOS-Chem at $2° \times 2.5°$ resolution using c360 meteorology. There is substantial improvement in the tropics relative to the standard GEOS-Chem simulation using archived convective mass fluxes (left panel). Extratropical regions show less improvement, as vertical transport is driven more by baroclinic instability rather than convection.

We further investigated whether spatial averaging of boundary layer mixing depths from the high-resolution meteorological archive could weaken vertical transport in coarse-resolution off-line simulations. As shown by the diagram in Figure 8, spatial averaging of mixing depths prevents boundary layer mixing to higher altitudes that would otherwise take place in part of the domain. This mixing would then drive a circulation ventilating a larger fraction of the domain to higher altitudes than specified by the average mixing depth. To assess the potential magnitude of this effect, we conducted a sensitivity simulation using the maximum $0.25° \times 0.3125°$ mixing depth within a $2° \times 2.5°$ grid cell as the mixing depth for that grid cell. These mixing depths are used in place of the mean mixing depths in boundary layer mixing and any other process that takes mixing depth as input (including RAS). The result is shown in the right panel of Figure 6. The simulation of surface concentrations is improved although there is overcompensation in the middle troposphere.

## 6  Conclusions

In this work, we isolated the different sources of transport errors resulting from performing off-line chemical transport model (CTM) simulations with archived meteorological data from a general circulation model (GCM). Errors include temporal averaging and remapping of the meteorological archive, differences in transport algorithms (sometimes required by lack of information in the archive), and coarsening of the CTM grid to enable simulations with a large number of chemically coupled species. We then explored some possibilities for reducing these errors.

We used as reference an on-line simulation of the $^{222}$Rn-$^{210}$Pb-$^{7}$Be chemical tracer suite in the Goddard Earth Observing System (GEOS-5) GCM at cubed-sphere c360 ($\approx$25 km) resolution. The operational meteorological archive from GEOS-5, stored as 3-h averages (1-h for mixing depths) and remapped onto a $0.25° \times 0.3125°$ ($\approx$25 km) rectilinear grid, provides the standard input to the GEOS-Chem CTM used by a large research community for atmospheric chemistry applications. These applications often degrade the meteorological archive to $2° \times 2.5°$ ($\approx$200 km) horizontal resolution (coarse-resolution GEOS-Chem) to make global chemical simulations computationally tractable. We conducted an ensemble of simulations to document the cascade of errors involved in going from the on-line GEOS-5 high-resolution simulation to the off-line GEOS-Chem coarse-resolution simulation. Although our study focuses on a particular GCM-CTM combination, our findings have relevance for any CTM driven by meteorology produced at high resolution. Vertical transport errors are of particular interest and the $^{222}$Rn-$^{210}$Pb-$^{7}$Be tracer suite is well-suited for that purpose.

We first diagnosed the cascade of transport errors in the $^{222}$Rn simulation when going from the on-line c360 GEOS-5 simulation to the off-line $0.25° \times 0.3125°$ GEOS-Chem simulation. The error from using temporally-averaged convective





mass fluxes is relatively small. The errors from using archived winds and from remapping of c360 fields to $0.25° \times 0.3125°$ are both more severe, resulting together in 5-20% biases. Transport from the boundary layer to the upper troposphere is too weak in the off-line model and this is due at least in part to loss of transient organized advective motions (resolved convection) in the 3-hour averaging of the meteorological archive.

5    We then examined the effect of degrading the spatial resolution of the meteorological archive from $0.25° \times 0.3125°$ to $2° \times 2.5°$ for input to the coarse-resolution GEOS-Chem. This further weakened vertical transport by up to 40% as organized vertical motions in the $0.25° \times 0.3125°$ archive were averaged out in the $2° \times 2.5°$ archive. The weakened vertical transport also affected by 5-10% the lifetimes of $^{210}$Pb and $^{7}$Be against deposition.

We explored different possibilities for restoring vertical transport in the off-line coarse-resolution simulation. Archiving 10    eddy vertical winds between adjacent vertical layers was found to be of negligible benefit, indicating that the loss of non-local organized vertical motions is more important. Spatial averaging of boundary layer mixing depths leads to underestimates of vertical transport and this can be corrected by weighting the averaging towards higher values. We showed that the loss of vertical organized convective motions could be corrected to some extent by using the on-line GCM convection scheme (here the Relaxed-Arakawa-Schubert or RAS) to operate at the coarse-resolution of the CTM using the meteorological archive as 15    input. This improves vertical transport in the tropics though has little effect at higher latitudes.

Our work has revealed significant vertical transport errors in off-line CTM applications when using meteorological archives from a GCM operating at high resolution. As the resolution of the GCMs continue to increase, the transport information lost in off-line CTMs will also increase. This may be corrected by increasing the frequency of archiving, avoiding remapping of the archive, using consistent transport algorithms, and applying scale-dependent convective transport parameterizations off-line. 20    We plan to include these improvements in future versions of the standard GEOS-Chem code.

**Code availability**

GEOS-Chem source code is freely available to the public. Source code may be downloaded by following instructions found at http://wiki.geos-chem.org/. At the time of writing, this work used a modified version of GEOS-Chem version 11-01 (the most recent public release) as indicated in the text. All developments presented here will be included in the next public release (ver- 25    sion 11-02) of GEOS-Chem (http://wiki.seas.harvard.edu/geos-chem/index.php/GEOS-Chem_v11-02). If you wish to access the code used in this work prior to the public release of version 11-02, you may do so at https://github.com/kyu0110/geoschem_ras.

*Acknowledgements.* This work was funded by the Modeling, Analysis, and Prediction (MAP) program of the NASA Earth Science Division.





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






**Figure 1.** Ensemble of global $^{222}$Rn-$^{210}$Pb-$^{7}$Be simulations conducted in this work. The blue and green boxes identify simulations originating from reference high-resolution (c360) and coarse-resolution (c48) GEOS-5 meteorological products, respectively.

Strode, S. A., Duncan, B. N., Yegorova, E. A., Kouatchou, J., Ziemke, J. R., and Douglass, A. R.: Implications of carbon monoxide bias for methane lifetime and atmospheric composition in chemistry climate models, Atmospheric Chemistry and Physics, p. 11789–11805, 2015.

5  Wang, Y. X., McElroy, M. B., Jacob, D. J., and Yantosca, R. M.: A nested grid formulation for chemical transport over Asia: Applications to CO, Journal of Geophysical Research: Atmospheres, 109, 2004.

Wu, S., Mickley, L. J., Jacob, D. J., Logan, J. A., Yantosca, R. M., and Rind, D.: Why are there large differences between models in global budgets of tropospheric ozone?, Journal of Geophysical Research: Atmospheres, 112, n/a–n/a, doi:10.1029/2006JD007801, http://dx.doi.org/10.1029/2006JD007801, d05302, 2007.

Zhang, B., Liu, H., Crawford, J. H., Fairlie, T. D., Chen, G., Dibb, J. E., Shah, V., Sulprizio, M., and B., Y.: Constraints from airborne $^{210}$Pb observations on aerosol scavenging and lifetime in a global chemical transport model, presented at 97th AMS Annual Meeting, 2017.



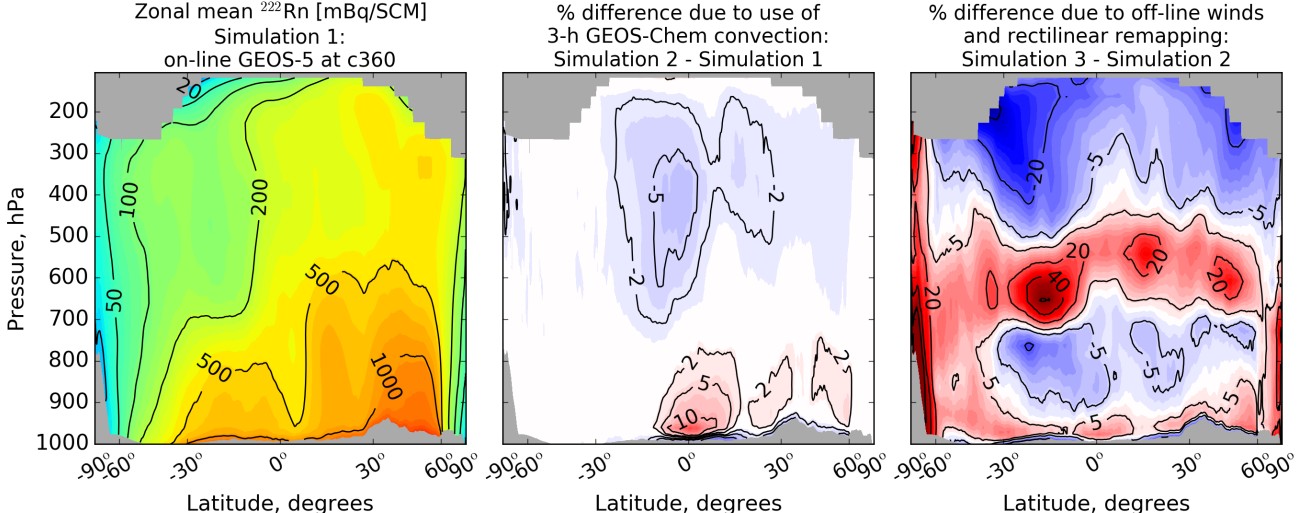

**Figure 2.** Off-line transport errors in $^{222}$Rn concentrations simulated at c360 ($\approx$25 km) resolution. Left panel: zonal mean concentrations of $^{222}$Rn from the on-line GEOS-5 reference simulation with c360 cubed-sphere resolution (Simulation 1). Values are monthly means for July 2013 after a 1-month spin-up from zero concentrations and are expressed in mixing ratio units of milli bequerels per standard cubic meter (at 0°C and 1 atm pressure) or mBq SCM$^{-1}$. Middle panel: errors due to the use of simplified off-line GEOS-Chem convection, shown as percentage differences between Simulation 2 and Simulation 1. Right panel: errors in the high-resolution GEOS-Chem simulation at $0.25° \times 0.3125°$ ($\approx$25 km) resolution due to off-line archiving of winds and mixing depths, remapping to rectilinear grid, and use of different transport schemes, shown as percentage differences between Simulation 3 and Simulation 2. The abscissa is on a sine latitude (equal area) scale. Stratospheric results are not shown (see text). Here and in other figures, solid color contours provide finer gradation of the labeled line contours.





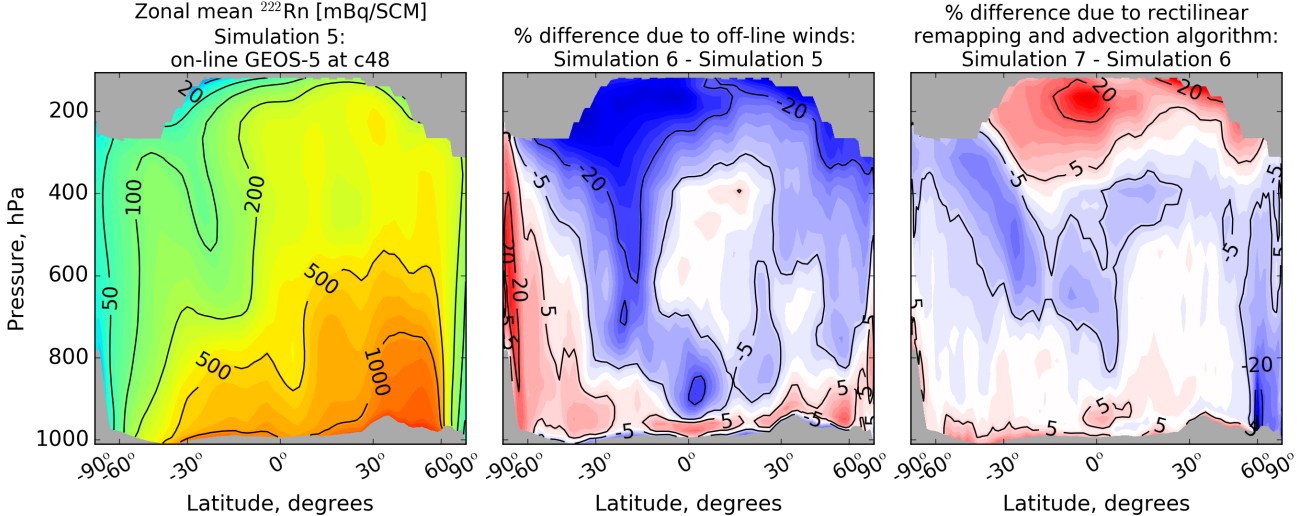

**Figure 3.** Off-line transport errors in $^{222}$Rn concentrations simulated at c48 ($\approx$200 km) resolution. Left panel: zonal monthly mean concentrations of $^{222}$Rn for July 2013 from the on-line GEOS-5 simulation with c48 cubed-sphere resolution (Simulation 5). Middle panel: errors due to off-line archiving of meteorological fields (no remapping), shown as percentage differences between Simulation 6 and Simulation 5. Right panel: errors due to remapping of the meteorological archive from c48 to $2° \times 2.5°$ (rectilinear, $\approx$200 km) and use of a lower-order advection scheme, shown as percentage differences between Simulation 7 and Simulation 6.

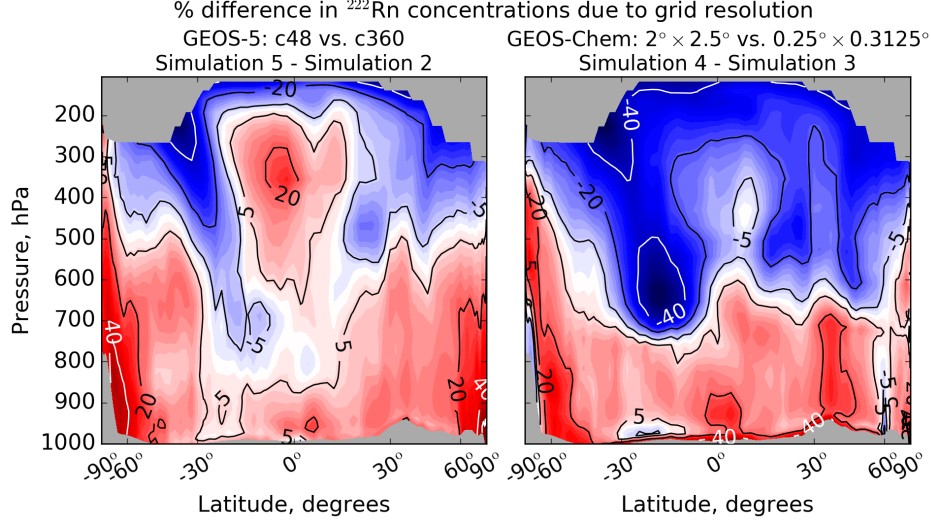

**Figure 4.** Effect of grid resolution on simulated zonal mean $^{222}$Rn concentrations for July 2013. Left: percentage differences between on-line GEOS-5 simulations at c48 resolution (Simulation 5) and c360 resolution (Simulation 2). Right: percentage differences between off-line GEOS-Chem simulations at $2° \times 2.5°$ resolution (Simulation 4) and $0.25° \times 0.3125°$ resolution (Simulation 3).




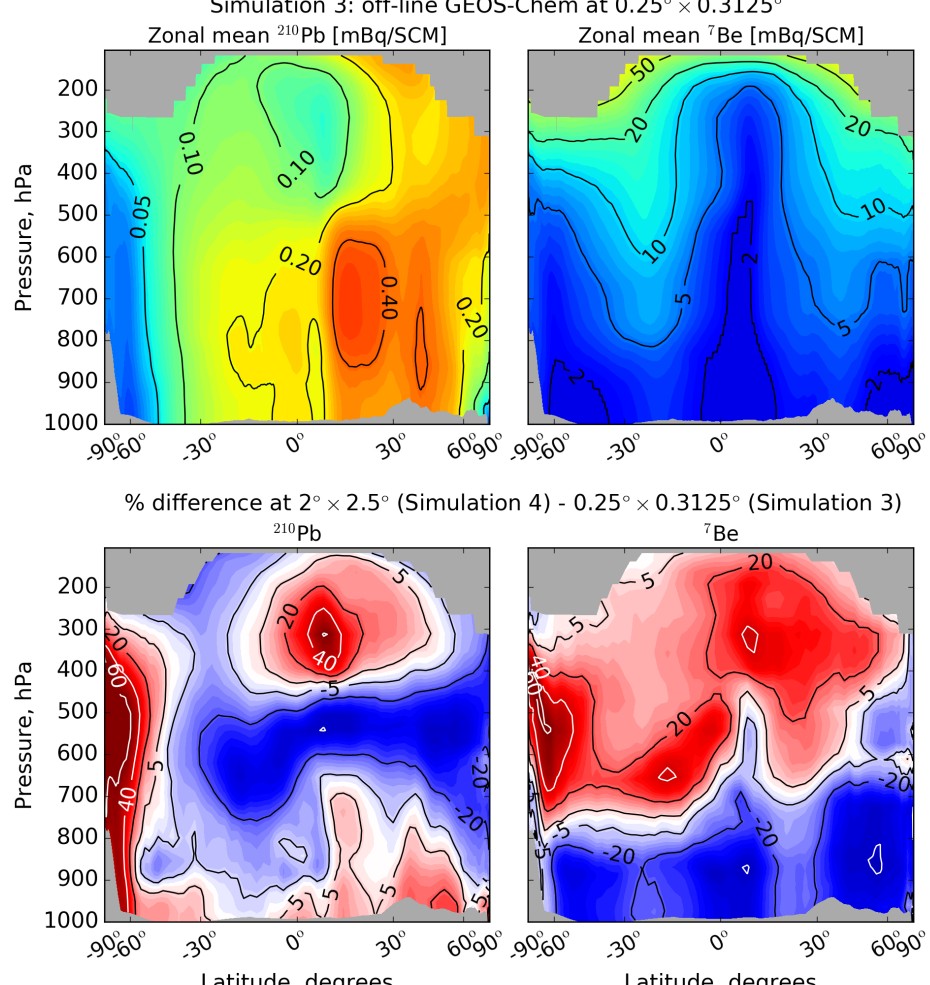

**Figure 5.** Simulated GEOS-Chem zonal mean concentrations of $^{210}$Pb and $^{7}$Be at $0.25° \times 0.3125°$ grid resolution (Simulation 3, top panels), and effect of degrading grid resolution to $2° \times 2.5°$ (Simulation 4, bottom panels). Concentrations are monthly means for July 2013 after a 1-month spin-up from zero concentrations and are expressed in mixing ratio units of milli bequerels per standard cubic meter (at $0°$C and 1 atm pressure) or mBq SCM$^{-1}$. The bottom panels show percentage differences between Simulation 4 and Simulation 3.



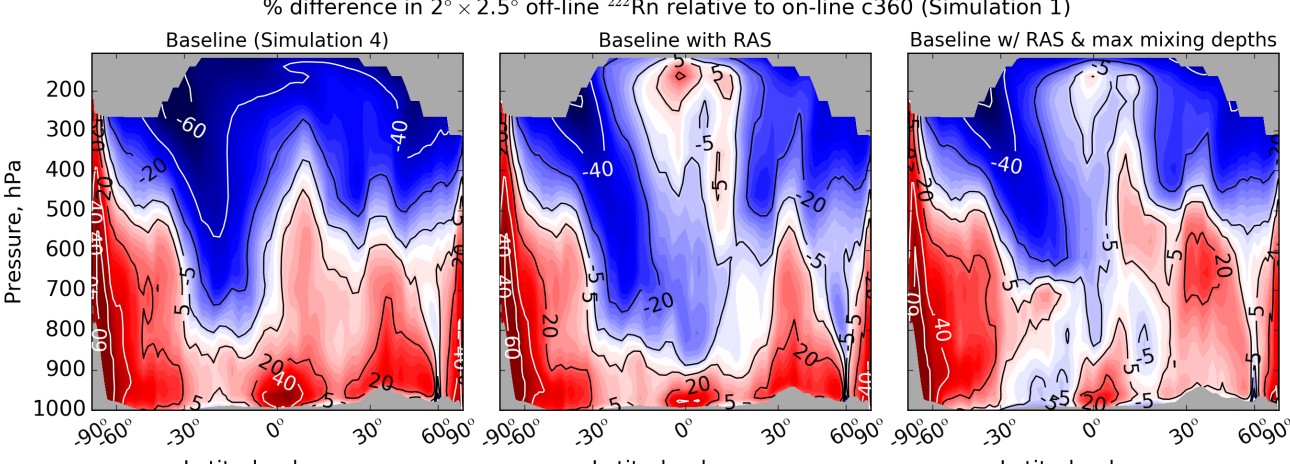

**Figure 6.** Errors in off-line coarse-resolution ($2° \times 2.5°$) simulations of $^{222}$Rn concentrations relative to the reference on-line c360 GEOS-5 simulation (Simulation 1). Values are percentage differences of zonal mean concentrations for July 2013. The left panel shows errors for the baseline $2° \times 2.5°$ GEOS-Chem simulation. The middle panel shows errors for the same baseline but with adjusted RAS convective mass fluxes (see text). The right panel shows errors for the same baseline with adjusted RAS convective mass fluxes and maximum mixing depths for each coarse-resolution grid cell (cf. Figure 8).

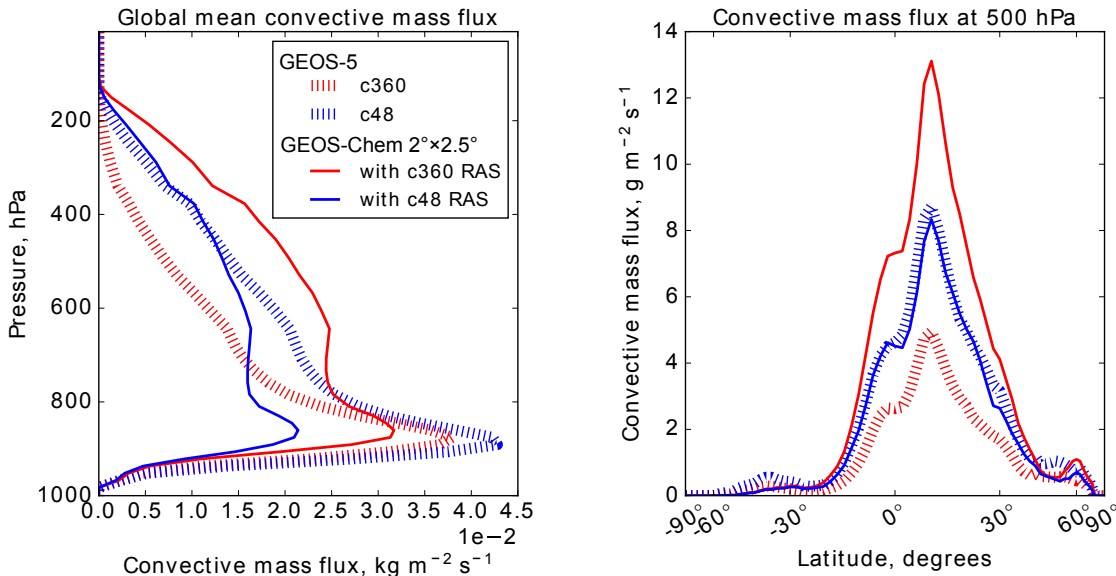

**Figure 7.** Convective mass fluxes for July 2013 produced by the GEOS-5 GCM at cubed-sphere c360 ($\approx$25 km) and c48 ($\approx$200 km) resolution using the Relaxed Arakawa Schubert (RAS) scheme, and by the $2° \times 2.5°$ ($\approx$200 km) GEOS-Chem simulation using the RAS scheme with c360 and c48 GEOS-5 meteorology. Left: vertical profile of global mean convective mass flux. Right: zonal mean convective mass flux at 500 hPa as a function of latitude.





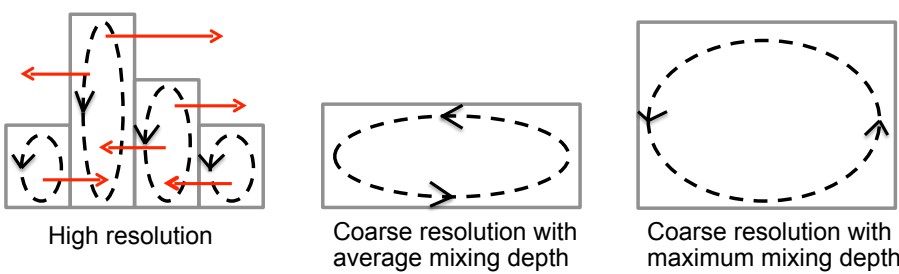

**Figure 8.** Effect of averaging high-resolution mixing depths in coarse-resolution simulations. The left panel illustrates mixing at high grid resolution, where the vertical extents of the boxes denote mixing depths and the dashed arrows illustrate the mixing. Red arrows show the induced circulation for a chemical tracer emitted at the surface. The middle and right panels illustrate mixing at coarse resolution with mixing depth taken as the average or the maximum of the high-resolution values.