# Peer review of "Errors and improvements in the use of archived meteorological data for chemical transport modeling: an analysis using GEOS-Chem v11-01 driven by GEOS-5 meteorology"

_Geoscientific Model Development, 2017_

## Referee Comment (RC1) · Anonymous Referee #1 · 18 Jul 2017

SUMMARY: Well-written systematic evaluation of the biases in trace gas simulations resulting from the choice to run the model off-line, run at a reduced resolution, or use a different coordinate system in the CTM than the GCM. The analysis shows that the differences are substantial at least at fairly high resolutions. The authors also suggest a couple of easy fixes that ameliorate the worst of the biases.

General Comments: A more informative title would be useful.

The cubed-grid coordinate system used in the GEOS GCM is relatively new. Are other GCMS adapting this scheme.? What other global CTMs if any are being affected by the transition of GCMs from rectilinear to cubed-grid coordinate systems?

Did this analysis reveal any surprising insights into the types of analyses that should

be performed with off-line CTMs versus on-line CTMs?

Minor comments:

P3L24: Off-line 4x5 global simulations are increasingly uncommon. Perhaps you should use 2x2.5 and 1x1 as the standards here.

P4L27: "Both versions use the same archived data". Is this true? I thought the data was archived on a rectilinear grid and not a cubed-sphere grid?

The C48 percent difference due to rectilinear mapping and the use of a lower order advection algorithm is shown in Figure 3c. Is this difference large enough to warrant archival of meteorological fields on the native cube-sphere grid? Why wasn't a high-performance GEOS-Chem cubed sphere calculation performed at C360?

P8L10: Is the bulk convective scheme used in GEOS-Chem also used in other off-line CTMs that use RAS?

P12L7-15: The use of maximum mixing depths instead of mean mixing depths is an interesting approach. What transport modules other than RAS use these mixing depths as input? Is maximum mixing depth currently being archived?

Figure 7: Could you explain why the C48 and GEOS-Chem 2x2.5 with C48 RAS convective mass flux distributions are nearly identical at 500 hPa but quite different at other altitudes.

P13L20: Could you provide a prioritized wish list for improvements? e.g., Should Lin and Rood be replaced by Putnam and Lin? ... and if yes, what are the implications for data storage etc.

PL1320: Is retaining the nested-grid capability of GEOS-Chem a priority? If yes, how would these improvements also help nested-grid simulations or could they cause problems?

---

## Short Comment (SC1) · 18 Jul 2017

Dear authors,

In my role as Executive editor of GMD, I would like to bring to your attention our Editorial version 1.1:

http://www.geosci-model-dev.net/8/3487/2015/gmd-8-3487-2015.html

This highlights some requirements of papers published in GMD, which is also available on the GMD website in the 'Manuscript Types' section:

http://www.geoscientific-model-development.net/submission/manuscript_types.html

In particular, please note that for your paper, the following requirements have not been

met in the Discussions paper:

- "The main paper must give the model name and version number (or other unique identifier) in the title."

- "If the model development relates to a single model then the model name and the version number must be included in the title of the paper. If the main intention of an article is to make a general (i.e. model independent) statement about the usefulness of a new development, but the usefulness is shown with the help of one specific model, the model name and version number must be stated in the title. The title could have a form such as, "Title outlining amazing generic advance: a case study with Model XXX (version Y)"."

- "All papers must include a section, at the end of the paper, entitled 'Code availability'. Here, either instructions for obtaining the code, or the reasons why the code is not available should be clearly stated. It is preferred for the code to be uploaded as a supplement or to be made available at a data repository with an associated DOI (digital object identifier) for the exact model version described in the paper. Alternatively, for established models, there may be an existing means of accessing the code through a particular system. In this case, there must exist a means of permanently accessing the precise model version described in the paper. In some cases, authors may prefer to put models on their own website, or to act as a point of contact for obtaining the code. Given the impermanence of websites and email addresses, this is not encouraged, and authors should consider improving the availability with a more permanent arrangement. After the paper is accepted the model archive should be updated to include a link to the GMD paper."

Therefore please change the title of your manuscript upon the revised submission to something like "Errors and improvements in the use of archived meteorological data for

chemical transport modeling: a case study with GEOS-5 GCM and GEOS-Chem CTM (v11_XX)

As webpages and github are no permanent archives, the executive Editors of GMD highly recommend to make the exact code version, to which the paper referes to, available via an archiving system providing a DOI (e.g. Zenodo). In this way, the code version and the paper are perfectly linked to each other.

A personal comment in the end: I really doubt the first sentence of your abstract. Look at the CCMI models (https://doi.org/10.5194/gmd-10-639-2017) and your can not hold your statement that "Global simulations of atmospheric chemistry are **generally** conducted with **off-line chemical transport models** (CTMs) [...]".

Yours,

Astrid Kerkweg

---

## Referee Comment (RC2) · Anonymous Referee #2 · 29 Sep 2017

This work addresses issues related to vertical transport in offline CTMs that are run at coarse resolutions. Because analysis or reanalysis meteorology is typically computed at much finer spatial and temporal scales, the use of these fields to drive offline fields can cause several problems. First, the use of 3D fields that are archived every 3 to 6 hours results in a loss of information aboute shorter time scale mixing processes, even when the CTM is run at the same resolution as the meteorological analysis. Second, averaging meteorological fields spatially to a much coarser grid reduces vertical mass flux because the role of subgrid transport processes differs substantially at fine and coarse resolutions and this difference is not accounted for in CTMs. These effects are examined using a combination of short-lived tracer experiments in the offline GEOS-Chem and an online version of the GEOS-5 GCM. This an important but previously

unaddressed issue that affects the use of CTMs for a number of scientific problems. The analysis is insightful, thorough, and well written. I recommend publication after the minor comments below are addressed.

General Comments:

The transport issues are well illustrated using the zonal mean plots of Rn, Be, and Pb. It would be interesting to see how large the horizontal variations were. For example, I would assume that the large mid- and upper troposphere differences seen in the left panel of Figure 6 are larger over land masses (where Rn is emitted) than over oceans. I don't think its necessary to add more figures, but it would be nice to see a discussion of the horizontal and temporal variability of these errors discussed somewhere in the text.

These issues would appear to be problematic for a wide range of longer lived gases that GEOS-Chem and other CTMs are used to study (e.g. CO, CH4). Could you comment on the implications for other species in the conclusions?

Technical Comments:

P2, line 8 – What version of the GEOS analyses are being used to drive GEOS-Chem?

P4, line 20 – This online capability is being used here as a comparison against the offline GEOS-Chem runs? Should mention this here.

P8, line 12 – Which way is GEOS-5's RAS transport done – short plume to tall or tall to short?

Figure 3 – Does the middle panel also include the difference in online vs offline PBL mixing? This is mentioned in the discussion of Figure 2 (p8, line 19) and appears to have a substantial impact in the left panel of that figure. However, it is not noted in figure 3 and its impact is not discussed.

P9, line 15 – Also worth noting that while the percentage change in the figure is large,

the absolute mixing ratio at the poles are quite low because of the short lifetime of Rn and lack of emission over ice/snow covered land masses. So this should be interpreted as very large uncertainty about a very small number.

P10, line 21 – The use of the term 'on-line archive' is confusing since the archive isn't used in the online simulation. Maybe this need to be made more explicit – e.g. 'archiving of fields in support of offline simulations'.

P10, line 27 – Is omega saved every 3 hours as is standard in GEOS? Or more frequently for these experiments?

---

## Author Comment (AC1) · 26 Oct 2017

We thank the topical editor and all reviewers for their feedback and suggestions. We address all reviewers' comments and have attached a revised the manuscript showing changes made based on the feedback.

**Short comment from executive editor A. Kerkweg:**

As per journal requirements, we have modified the title of our paper to include the model version. The title is now "Errors and improvements in the use of archived mete-orological data for chemical transport modeling: an analysis using GEOS-Chem v11-

01 driven by GEOS-5 meteorology". We have also uploaded the code to Zenodo as recommended and associated the DOI of the publication (https://doi.org/10.5194/gmd-2017-125) with the code, although the best place to obtain up-to-date information about GEOS-Chem is still the website.

*A personal comment in the end: I really doubt the first sentence of your abstract. Look at the CCMI models (https://doi.org/10.5194/gmd-10-639-2017) and your can not hold your statement that "Global simulations of atmospheric chemistry are generally conducted with off-line chemical transport models (CTMs) [...]".*

We've changed "generally" to "commonly". Although there has been much progress in chemistry-climate models, there are still many in the atmospheric chemistry community who rely on offline CTMs.

**Referee Comment 1:**

*A more informative title would be useful.*

We have modified our title to "Errors and improvements in the use of archived meteorological data for chemical transport modeling: an analysis using GEOS-Chem v11-01 driven by GEOS-5 meteorology", which we hope readers will find more informative.

*The cubed-grid coordinate system used in the GEOS GCM is relatively new. Are other GCMS adapting this scheme.? What other global CTMs if any are being affected by the transition of GCMs from rectilinear to cubed-grid coordinate systems?*

We have added the following sentences: "The cubed-sphere grid has been used in other GCMs, such as the GFDL AM3 (Donner et al., 2011). " "Although we use GEOS-Chem in our comparisons against on-line GEOS-5 GCM results, the issues discussed in this paper are more generally pertinent to CTMs driven by archived GCM meteorological data."

[Figure]

*Did this analysis reveal any surprising insights into the types of analyses that should be performed with off-line CTMs versus on-line CTMs?*

We have added to the conclusions: "Given these large differences in vertical transport, users examining the effect of convection on a chemical species should take care to perform their simulations at sufficiently high resolution. Those conducting simulations of long-lived trace gases such as CO2 or CH4 should also be aware of these errors."

*P3L24: Off-line 4x5 global simulations are increasingly uncommon. Perhaps youshould use 2x2.5 and 1x1 as the standards here.*

We agree that $4° \times 5°$ global simulations are increasingly uncommon. Since all our comparisons are against $2° \times 2.5°$, references to $4° \times 5°$ in the text are unnecessary and we have removed them.

*P4L27: "Both versions use the same archived data". Is this true? I thought the data was archived on a rectilinear grid and not a cubed-sphere grid?*

For this study, we archived data on the cubed-sphere grid, and regridded the data to rectilinear for use in GEOS-Chem. We revised that sentence to read "Both versions use the same modules except for advection." to avoid confusion.

*The C48 percent difference due to rectilinear mapping and the use of a lower order advection algorithm is shown in Figure 3c. Is this difference large enough to warrant archival of meteorological fields on the native cube-sphere grid? Why wasn't a highperformance GEOS-Chem cubed sphere calculation performed at C360?*

Yes, we suggest archival of meteorology on the cubed-sphere grid as an improvement in section 5. We have updated the conclusions to make to this point clearer. We have added to section 3.2: The c48 resolution allowed us to conduct a cubed-sphere off-line (GCHP) simulation, which we were not able to do at c360 resolution due to computational limitations.

*P8L10: Is the bulk convective scheme used in GEOS-Chem also used in other off-line CTMs that use RAS?*

We have added the following sentence to the paragraph: "Bulk convective transport using archived mass fluxes is a standard procedure in other CTMs such as TOMCAT (Feng et al., 2011)."

*P12L7-15: The use of maximum mixing depths instead of mean mixing depths is an interesting approach. What transport modules other than RAS use these mixing depths as input? Is maximum mixing depth currently being archived?*

We've edited the text to make it clearer that the maximum mixing depths are used in boundary layer mixing and in RAS. This is not currently archived in the standard GEOS-5 meteorology.

*Figure 7: Could you explain why the C48 and GEOS-Chem 2x2.5 with C48 RAS convective mass flux distributions are nearly identical at 500 hPa but quite different at other altitudes.*

We have added the following to section 5: "The larger difference between GEOS-5 and GEOS-Chem convective mass fluxes below 500 hPa compared to above 500 hPa suggests that convective motions penetrating higher altitudes are more likely to be retained after temporal averaging."

*P13L20: Could you provide a prioritized wish list for improvements? e.g., Should Lin and Rood be replaced by Putnam and Lin? ... and if yes, what are the implications for data storage etc.*

We have rearranged our suggestions in the conclusions in order of priority and added a sentence on implications for data storage and computational power. The section now reads: "As the resolution of the GCMs continue to increase, the transport information lost in off-line CTMs will also increase. This may be corrected, in order of priority, by 1) applying scale-dependent convective transport parameterizations off-line, 2) avoiding remapping of the archive by archiving on the cubed-sphere grid, 3) using consistent transport algorithms (in the case of GEOS-Chem, Putman and Lin, 2010 rather than Lin and Rood, 1996), and 4) increasing the frequency of archiving. Of the list, 1) will only require a minor increase in computational time, 2) and 4) will increase both data storage and computational resources, and 3) will require no additional resources if 2) is

done. These improvements will benefit off-line simulations at all resolutions, including high-resolution nested simulations. We plan to include these improvements in future versions of the standard GEOS-Chem code."

*PL1320: Is retaining the nested-grid capability of GEOS-Chem a priority? If yes, how would these improvements also help nested-grid simulations or could they cause problems?*

We have added to the conclusions: "These improvements will benefit off-line simulations at all resolutions, including high-resolution nested simulations. " However, nested simulations are not currently a capability in the cubed-sphere version of GEOS-Chem, and development priorities are adapted based on the needs of the community.

**Referee Comment 2:**

*The transport issues are well illustrated using the zonal mean plots of Rn, Be, and Pb. It would be interesting to see how large the horizontal variations were. For example, I would assume that the large mid- and upper troposphere differences seen in the left panel of Figure 6 are larger over land masses (where Rn is emitted) than over oceans. I don't think its necessary to add more figures, but it would be nice to see a discussion of the horizontal and temporal variability of these errors discussed somewhere in the text.*

Due to resource limitations, we conducted our simulation for only one month, making it difficult to comment on temporal variability.

*These issues would appear to be problematic for a wide range of longer lived gases that GEOS-Chem and other CTMs are used to study (e.g. CO, CH4). Could you comment on the implications for other species in the conclusions?*

We have added the following to the conclusions: "Given these large differences in vertical transport, users examining the effect of convection on a chemical species should take care to perform their simulations at sufficiently high resolution. Those conducting simulations of long-lived trace gases such as $CO_2$ or $CH_4$ should also be aware of

these errors."

*P2, line 8 – What version of the GEOS analyses are being used to drive GEOS-Chem?*

We clarify in the text that we use a custom product.

*P4, line 20 – This online capability is being used here as a comparison against the offline GEOS-Chem runs? Should mention this here.*

We have added the following sentence: Although we use GEOS-Chem in our comparisons against on-line GEOS-5 GCM results, the issues discussed in this paper are more generally pertinent to CTMs driven by archived GCM meteorological data.

*P8, line 12 – Which way is GEOS-5's RAS transport done – short plume to tall or tall to short?*

We modified the sentence to read: "One explanation for why a multi-plume parameterization might produce a different transport pattern is that each sequential plume acts on a different concentration gradient that has been modified by the previous plume, until the moisture and temperature fields are balanced."

*Figure 3 – Does the middle panel also include the difference in online vs offline PBL mixing? This is mentioned in the discussion of Figure 2 (p8, line 19) and appears to have a substantial impact in the left panel of that figure. However, it is not noted in figure 3 and its impact is not discussed.*

We have changed the label on Figure 3 and edited the text to make it more clear that it is the effect of having all meteorological fields offline, not just the winds.

*P9, line 15 – Also worth noting that while the percentage change in the figure is large, the absolute mixing ratio at the poles are quite low because of the short lifetime of Rn and lack of emission over ice/snow covered land masses. So this should be interpreted as very large uncertainty about a very small number.*

We have added the following sentence: While these differences are large, absolute concentrations are very low over the poles due to the short lifetime of 222Rn and lack of emission over ice and snow.

*P10, line 21 – The use of the term 'on-line archive' is confusing since the archive isn't used in the online simulation. Maybe this need to be made more explicit – e.g. 'archiving of fields in support of offline simulations'.*

We have corrected "on-line" to "off-line".

*P10, line 27 – Is omega saved every 3 hours as is standard in GEOS? Or more frequently for these experiments?*

We used the standard 3 hour values, and have made this clearer in the text.

Please also note the supplement to this comment:
https://www.geosci-model-dev-discuss.net/gmd-2017-125/gmd-2017-125-AC1-supplement.pdf

**Supplement:**

[revised manuscript text omitted]

% difference in 2° × 2.5° off-line ²²²Rn relative to on-line c360 (Simulation 1)

**Figure 6.** Errors in off-line coarse-resolution (2° × 2.5°) simulations of ²²²Rn concentrations relative to the reference on-line c360 GEOS-5 simulation (Simulation 1). Values are percentage differences of zonal mean concentrations for July 2013. The left panel shows errors for the baseline 2° × 2.5° GEOS-Chem simulation. The middle panel shows errors for the same baseline but with adjusted RAS convective mass fluxes (see text). The right panel shows errors for the same baseline with adjusted RAS convective mass fluxes and maximum mixing depths for each coarse-resolution grid cell (cf. Figure 8).

[Figure]

**Figure 7.** Convective mass fluxes for July 2013 produced by the GEOS-5 GCM at cubed-sphere c360 (≈25 km) and c48 (≈200 km) resolution using the Relaxed Arakawa Schubert (RAS) scheme, and by the 2° × 2.5° (≈200 km) GEOS-Chem simulation using the RAS scheme with c360 and c48 GEOS-5 meteorology. Left: vertical profile of global mean convective mass flux. Right: zonal mean convective mass flux at 500 hPa as a function of latitude.

[Figure]

**Figure 8.** Effect of averaging high-resolution mixing depths in coarse-resolution simulations. The left panel illustrates mixing at high grid resolution, where the vertical extents of the boxes denote mixing depths and the dashed arrows illustrate the mixing. Red arrows show the induced circulation for a chemical tracer emitted at the surface. The middle and right panels illustrate mixing at coarse resolution with mixing depth taken as the average or the maximum of the high-resolution values.

---

## Author Response (AR2)

1. Done

[revised manuscript text omitted]